# Decision-Theoretic Planning with Communication in Open Multiagent Systems

**Anirudh Kakarlapudi**[1]     **Gayathri Anil**[1]     **Adam Eck**[2]     **Prashant Doshi**[1]     **Leen-Kiat Soh**[3]

[1]Computer Science Department, University of Georgia, Athens, Georgia, USA
[2]Computer Science Department, Oberlin College, Oberlin, Ohio, USA
[3]School of Computing, University of Nebraska, Lincoln, Nebraska, USA

## Abstract

In open multiagent systems, the set of agents operating in the environment changes over time and in ways that are nontrivial to predict. For example, if collaborative robots were tasked with fighting wildfires, they may run out of suppressants and be temporarily unavailable to assist their peers. Because an agent's optimal action depends on the actions of others, each agent must not only predict the actions of its peers, but, before that, reason whether they are even present to perform an action. Addressing openness thus requires agents to model each other's presence, which can be enhanced through agents communicating about their presence in the environment. At the same time, communicative acts can also incur costs (e.g., consuming limited bandwidth), and thus an agent must tradeoff the benefits of enhanced coordination with the costs of communication. We present a new principled, decision-theoretic method in the context provided by the recent communicative interactive POMDP framework for planning in open agent settings that balances this tradeoff. Simulations of multiagent wildfire suppression problems demonstrate how communication can improve planning in open agent environments, as well as how agents tradeoff the benefits and costs of communication under different scenarios.

## 1 INTRODUCTION

When operating in a multiagent system, an optimizing agent benefits from reasoning about how other agents will behave–i.e., peer modeling–while choosing actions that maximize its chances of accomplishing shared or self-interested goals. However, nuances in real-world environments often introduce numerous sources of uncertainty that make such peer modeling challenging. One of these is **agent openness** that occurs whenever individual agents join or leave the system (temporarily or permanently) over time. For example, cooperative robots tasked with suppressing wildfires alongside or in place of human firefighters would need to periodically leave the environment to recharge their limited suppressants that were spent during firefighting. Likewise, competitive autonomous ride-sharing cars can no longer compete for new passengers while transporting a full ride. Consequently, openness requires that an agent not only predict *what* actions their neighbors will take, but also *whether* they are even present to take actions.

However, the presence or absence of neighbors is commonly unobservable to the optimizing agent. Instead, the agent is required to *infer* the dynamics of the agent population by the changes in the environment state. For instance, in the wildfire example, if a fire's intensity rises when the agent predicted it would decrease, then it's likely that neighbors were not present to help fight this fire. Chandrasekaran et al. [2016] introduced a decision-theoretic solution to this problem based on modeling the optimizing agent's decision problem as an IPOMDP-Lite [Hoang and Low, 2013]. Notably, in this solution, agents are not assumed to coordinate their behaviors or communicate information, making it applicable in a wide range of cooperative (e.g., wildfire suppression), competitive (e.g., autonomous ride-sharing), and self-interested scenarios, as well as ad hoc environments [Stone et al., 2010, Rahman et al., 2021, Mirsky et al., 2022].

On the other hand, if agents were instead capable of communicating with one another, then they could share information about their presence. That is, communicative acts, unlike regular actions that directly impact the physical state, can influence facets of the interacting agent's mental state such as its belief. Consequently, deciding to *purposefully* communicate requires modeling others' mental states and how communicative acts could change the receiving agent's belief, and subsequently its action. In this context, a recent framework called the communicative interactive POMDP (CI-POMDP) [Gmytrasiewicz and Adhikari,

*Accepted for the 38th Conference on Uncertainty in Artificial Intelligence* (UAI 2022).

2019, Gmytrasiewicz, 2020], building on the well-known I-POMDP [Gmytrasiewicz and Doshi, 2005], includes communicative acts, which leverages the framework's unique capability of modeling other agents' mental states, how they may change with time, and their modeling of others as a part of sequential decision making. To date, research into the CI-POMDP model [Gmytrasiewicz, 2020] has focused on the underlying mathematical model and exploring how agents would decide to communicate on the benchmark multiagent Tiger problem, but it has not been used to solve real-world challenges such as communicating in open agent systems.

In this paper, we present a new method for decision-theoretic planning that extends Gmytrasiewicz [2020] by leveraging the CI-POMDP as a point of departure to enable agents to plan with *both physical and communicative actions in open multiagent systems*. We make the following contributions:

1. We extend the CI-POMDP to *model the presence of other agents*, suitable for open agent systems, building on prior work on decision-theoretic planning in open environments [Chandrasekaran et al., 2016, Eck et al., 2020]. This represents the first decision-theoretic approach for using communication to mitigate the challenges posed by agent openness. Notably, in such open environments, agents benefit from a different vocabulary of communicative messages than in the original CI-POMDP, which also changes how messages are incorporated into the receiving agents' belief update to help address the challenge of motivating communication in hierarchical reasoning.

2. We present the CI-POMCP-PF$_O$ algorithm for online planning with the CI-POMDP in open agent systems. We also extend ideas in Monte Carlo Tree Search (MCTS) from single agent planning in problems with large observation spaces [Sunberg and Kochenderfer, 2017, Garg et al., 2019, Thomas et al., 2020] to multiagent planning. We expect this general planning algorithm to significantly expand the use of the CI-POMDP. It is also the first fully online method for reasoning about agent openness.

3. We conduct experiments in the benchmark wildfire suppression domain [Chandrasekaran et al., 2016] to investigate the impact of communication on agent behaviors and task accomplishment, as well as how the extended CI-POMDP model balances the cost-benefit of planning communication. The CI-POMCP-PF$_O$ algorithm led to statistically significantly higher rewards from increased task accomplishment through improved coordination of agent behaviors in several situations, and agents were able to flexibly reduce communication as costs increased while maintaining the *benefits* of communicating.

## 2 BACKGROUND

The communicative interactive POMDP (CI-POMDP) [Gmytrasiewicz, 2020] builds on the well-known finitely-nested I-POMDP framework to include

an additional action of sending a message, an additional observation of receiving messages, and a set of messages that are sent or received. Formally,

$$\text{CI-POMDP}_{i,l} \triangleq \langle Ag, IS_{i,l}, A, \Omega_i, M, T_i, O_i, R_i, \gamma, b_{i,l}^0 \rangle$$

- $Ag$ is a finite set of agents, which includes a *subject agent* $i$ whose decision making is modeled by the CI-POMDP to decide how to act and communicate with the other agents $j \in N(i)$ in its neighborhood $N(i) = Ag \setminus \{i\}$.

- $IS_{i,l}$ is the set of level $l$ interactive states, $IS_{i,l} = S \times \bigtimes_{j \in N(i)} \mathcal{M}_{j,l-1}$ for $l > 0$. Here, $S$ is the set of states of the decision-making problem, possibly factored into variables $\dot{S}_1 \times \dot{S}_2 \times \ldots \times \dot{S}_k$, such as the intensities of the $k$ wildfires in the problem. Each agent $j \in N(i)$ is ascribed a computable model from the set $\mathcal{M}_{j,l-1}$, $\theta_{j,l-1} = \langle b_{j,l-1}, \hat{\theta}_j \rangle$ where $b_{j,l-1}$ is the agent's belief over its level $l-1$ interactive state and $\hat{\theta}_j$ denotes the agent's frame. A frame represents the agent's capabilities and preferences. The level-0 interactive states $IS_{i,0} = S$. [1].

- $A = A_i \times \bigtimes_{j \in N(i)} A_j$ is the set of possible joint actions of the agents; e.g., each agent choosing to fight or not the fires in its neighborhood. For notational convenience, $\mathbf{a_{-i}} \in \bigtimes_{j \in N(i)} A_j$ denotes the joint action by agents in $N(i)$.

- $\Omega_i$ is the set of observations of agent $i$.

- $M$ is the set of messages sent and received by an agent. Let $m_{i \to j} \in M$ denote a message that is sent to an agent $j$ and $m_{i \leftarrow j} \in M$ denote a message that is received from $j$. Let $\mathbf{m_{i \leftarrow -i}}$ denote the vector of messages received by $i$ from to all other agents.

- $T_i(s, a_i, \mathbf{a_{-i}}, s') = P(s'|s, a_i, \mathbf{a_{-i}})$ gives the probabilities of stochastic state transitions caused by actions of $Ag$.

- $O_i(s', a_i, \mathbf{a_{-i}}, o_i) = P(o_i|a_i, \mathbf{a_{-i}}, s')$ models the probabilities of stochastic observations revealed to subject agent $i$ after joint action $(a_i, \mathbf{a_{-i}})$.

- $R_i(s, a_i, \mathbf{a_{-i}}, m_{i \to -i}) \in \mathbb{R}$ is the reward function of agent $i$ dependent on the state, joint actions, and messages sent to others. While there is a cost of sending messages, there is no cost to receiving (and processing) messages.

- $\gamma \in (0, 1]$ and $b_{i,l}^0$ are the discount factor and the initial belief state of subject agent $i$ over its level-$l$ interactive state space, respectively.

An agent with level $l > 0$ in the CI-POMDP framework updates its belief on performing an action and possibly sending

---

[1] Choosing an appropriate level of hierarchical reasoning depends on the application. Whereas level 0 is similar to single agent reasoning, higher levels represent greater strategic awareness of neighbors and their action impact on the environment. However, higher-level reasoning adds computational complexity (often exponential in $l$), and $l = 1$ or $2$ are common [Doshi and Gmytrasiewicz, 2009]

a message at the previous time step followed by receiving an observation and possibly a vector of messages at the current time step. The belief update shown below yields the new belief $b_{i,l}^t = Pr(IS_{i,l}^t|b_{i,l}^{t-1}, a_i^{t-1}, \mathbf{m}_{i\to-i}^{t-1}, o_i^t, \mathbf{m}_{i\leftarrow-i}^t)$:

$$
\begin{aligned}
b_{i,l}^t(is^t) = {}& \alpha \sum_{is^{t-1}} b_{i,l}(is^{t-1}) \\
& \times \prod_{j\in Ag/\{i\}} \left( \sum_{a_j^{t-1}} Pr(a_j^{t-1}, m_{j\to i}^{t-1}|\theta_{j,l-1}^{t-1}) \right) \\
& \times T_i(s^{t-1}, a_i^{t-1}, \mathbf{a}_{-i}^{t-1}, s^t) O_i(s^t, a_i^{t-1}, \mathbf{a}_{-i}^{t-1}, o_i^t) \\
& \times \prod_{j\in Ag/\{i\}} \left( \sum_{o_j^t} \tau_{\hat\theta_j}(b_{j,l-1}^{t-1}, a_j^{t-1}, m_{j\to i}^{t-1}, o_j^t, m_{j\leftarrow i}^t, b_{j,l-1}^t) \right. \\
& \left. \times O_j(s^t, a_j^{t-1}, \mathbf{a}_{-j}^{t-1}, o_j^t) \right).
\end{aligned}
\tag{1}
$$

Here, $m_{j\to i}^{t-1}$ is the message sent by agent $j$ to $i$ at timestep $t-1$, which is same as the message received by agent $i$ from $j$ at timestep $t$, $m_{i\leftarrow j}^t$, since the framework assumes a perfect communication channel. Thus, the term $Pr(a_j^{t-1}, m_{j\to i}^{t-1}|\theta_{j,l-1}^{t-1})$ makes those models of $j$ that support sending this message more probable. $\tau_{\hat\theta_j}(b_{j,l-1}^{t-1}, a_j^{t-1}, m_{j\to i}^{t-1}, o_j^t, m_{j\leftarrow i}^t, b_{j,l-1}^t)$ is 1 if agent $j$'s belief in $is^{t-1}$ updates to $b_{j,l-1}^t$ in $is^t$ upon performing its predicted action $a_j^{t-1}$ and sending message $m_{j\to i}^{t-1}$ to $i$ followed by receiving possible observation $o_j^{t-1}$ and $i$'s sent message $m_{j\leftarrow i}^t$ to $j$. A level-0 agent updates its belief using the POMDP belief update by first marginalizing the other agent from the transition and observation functions using a fixed probability distribution.

Analogously to I-POMDPs, subject agent $i$ assigns a value to each level $l$ belief, which is the expected cumulative, discounted rewards over a finite (or infinite) horizon $H$, $r_0 + \gamma r_1 + \gamma^2 r_2 + \ldots + \gamma^{H-1} r_{H-1}$, by maximizing the Bellman equation for each belief and action-message pair:

$$
\begin{aligned}
Q_{i,l}(b_{i,l}^t, a_i^t, m_{i\to-i}^t) = {}& \rho_i(b_{i,l}^t, a_i^t, m_{i\to-i}^t) \\
& + \gamma \sum_{o_i^{t+1}, \mathbf{m}_{i\leftarrow-i}^{t+1}} Pr(o_i^{t+1}, \mathbf{m}_{i\leftarrow-i}^{t+1}|b_{i,l}^t, a_i^t, m_{i\to-i}^t) \\
& \times V_{i,l}^{t+1}(b_{i,l}^{t+1})
\end{aligned}
\tag{2}
$$

$$
V_{i,l}^t(b_{i,l}^t) = \max_{a_i\in A_i, m_{i\to-i}^t} Q_i^t(b_{i,l}^t, a_i^t, m_{i\to-i}^t)
\tag{3}
$$

where

$$
\begin{aligned}
\rho_i(b_{i,l}^t, a_i^t, m_{i\to-i}^t) = {}& \sum_{is^t\in IS_{i,l}^t} b_{i,l-1}^t(is^t) \sum_{\mathbf{a}_{-i}\in A_{-i}} \\
& \times \prod_{j\in Ag} \sum_{m_{j\to-j}^t} Pr(a_j^t, m_{j\to-j}^t|\theta_{j,l-1}^t) R_i(s, a_i, \mathbf{a}_{-i}, m_{i\to-i}^t)
\end{aligned}
$$

and $b_{i,l}^{t+1}$ is the updated belief on performing action $a_i^t$ and sending message $m_{i\to-i}^t$ followed by receiving observation $o_i^{t+1}$ and messages $\mathbf{m}_{i\leftarrow-i}^{t+1}$.

Policy $\pi_{i,l}$ is then the distribution of those action and message pairs that maximize the Q-value:

$$
OPT(b_{i,l}^t) = \arg \max_{a_i\in A, m\in M} Q_{i,l}(b_{i,l}^t, a_i, m_{i\to-i})
\tag{4}
$$

$$
\pi_{i,l}(a_i, m_{i\to-i}|b_{i,l}^t) = \frac{1}{|OPT|} \quad \forall (a_i, m_{i\to-i}) \in OPT
\tag{5}
$$

Prior research [Gmytrasiewicz and Adhikari, 2019, Gmytrasiewicz, 2020] has used the CI-POMDP model to analyze the expected agent behaviors in the two-agent instance of the multiagent Tiger problem [Gmytrasiewicz and Doshi, 2005]. In our knowledge, no algorithm has been presented to solve the CI-POMDP for $OPT$ and $\pi$. We contribute such an algorithm in Sec. 4 that could be generally used for CI-POMDP, and particularly used toward reasoning about agent openness, as described in the next section.

## 3 PLANNING WITH COMMUNICATION IN OPEN AGENT SYSTEMS

We describe how agent openness has previously been modeled in decision-theoretic planning and how communication can enhance inference in such reasoning. But, the latter requires addressing the challenge of motivating communication in the context of nested modeling due to the hierarchical reasoning in the CI-POMDP.

### 3.1 MODELING OPEN AGENT SYSTEMS

In this paper, we focus on open systems where agents may leave the environment at any time and possibly reenter, but new agents do not join. [2] Nonetheless, this brings unique conceptual and computational challenges to planning.

In open systems, individual planning is complicated by the need of each agent to track which other agents are currently present in the system and to reason about the actions of those agents only. In wildfire suppression, each firefighter must know how many others are currently unavailable because they are recharging their suppressant, in order to focus on the behaviors of those currently fighting the fires. Note that a firefighter $j$'s absence is not the same as $j$ choosing to do nothing, and thus would lead to $i$ updating its beliefs differently when modeling $j$.

Prior research studying decision-theoretic planning in open environments has kept track of this information in two ways:

---

[2] We suggest how to relax this assumption in Sec. 6.

either by maintaining coalitions of operating agents in the Open Dec-POMDP [Cohen et al., 2017], or by adding a presence state variable $present_j$ for each agent $j \in N(i)$ that indicates the neighbor's current presence in the system [Chandrasekaran et al., 2016]. Eck et al. [2020] proposed moving the $present_j$ state variables from the environment state $s$ into the mental models $\mathcal{M}_{j,l-1}$ maintained by the subject agent in an IPOMDP-Lite model [Hoang and Low, 2013] of the environment in order to gain efficiencies in the problem state space. We adopt this latter approach.

However, the presence or absence of other agents may not be directly observed by the subject agent in practice. On the other hand, it may infer the neighboring agents' absences from the interaction through sensing a lack of expected change in the state variables. For example, if firefighting agent $i$ senses that the intensity of a shared large fire is not reducing despite fighting the fire, it likely infers that its neighbors are not assisting. If agent $i$ believes that their suppressant levels were previously low, it may further infer that those agents are currently not participating in the interaction. However, this indirect inference is slow, unreliable, and often post hoc. We address this weakness of existing open agent reasoning through communication.

### 3.2 COMMUNICATION TO AID INFERENCE

Of course, a direct communication modality between agents may yield faster inference about the presence of other agents. To illustrate, a neighboring agent that sends the message that its suppressant level is very low shares a piece of information that enables others to predict that it will likely be absent from the interaction in the next time step. The planning agent can then update its belief to give a higher probability mass to those models of the other agent that have the $present_j$ variable as false. This enables the planning agent to have more informed beliefs about the presence of its neighbors and act more quickly to changes in the environment.

These $present_j$ variables represent the significant information required to address the challenges of reasoning about neighbor behavior in open agent systems, but cannot be *observed directly* by the subject agent. Therefore, allowing agents to communicate related to their presence makes available information that is otherwise unobservable and should improve the modeling by neighbors. For instance, if all agents now ascribe high beliefs to the presence or absence of the same agents, they are likely to coordinate better after reducing uncertainty in the important presence states. Agents may choose to share such private information if the resulting changes to neighbors' behaviors are beneficial to itself, such as a more coordinated use of the limited suppressant during wildfire suppression.

Previously, agents in a CI-POMDP communicated messages that represent marginals over their beliefs about the environment state as a way of affecting the beliefs of their neighbors [Gmytrasiewicz, 2020]. Instead, in open agent systems, let the set of messages $M$ that are sent and received by agents relate to the agent exiting the interaction or reentering it. In our firefighting example, we may let $M = \{$'*Have suppressant*', '*Nearly out of suppressant*', '*Out of suppressant*', '*No message*'$\}$. All of these messages pertain to the suppressant level of the communicating agent, which determines whether the agent is able to currently participate in the interaction or not. Messages in the CI-POMDP framework reveal information about the sender's mental models as shown in the belief update (Eq. 1). Therefore, the example messages in $M$ allow the receiver to update its belief about the sender being present or absent from the firefighting.

Henceforth, we denote a CI-POMDP that models the presence of agents and communicates about presence states as the *open agent CI-POMDP*, enabling decision-theoretic reasoning about not only how agents should act, but *when and what they should communicate*, in open agent systems.

### 3.3 NESTED MODELING COMPLICATES COMMUNICATION

Although CI-POMDPs offer a way to integrate communicative acts into decision making, a challenge is that the nested modeling of others as practiced in the framework inhibits communication. To understand this, recall that $IS_{i,0} = S$. In other words, level-0 agents in I-POMDPs do not ascribe intentional models to others in their environment. Instead, they may ascribe a flat probability distribution to the predicted actions of others that facilitates marginalizing others' actions, or ignoring others' presence. Subsequently, messages received from others may not influence a level-0 agent's beliefs. An unintended consequence of this is that the level-1 agent may decide to not communicate with its neighbor because it reasons that any message it sends may not influence the neighbors's level-0 belief. In our wildfire suppression example, a level-1 agent may not choose to communicate its suppressant level because it does not believe that its level-0 neighbors can make use of such information; instead, communicating would only incur a cost with no benefit through affected neighbor behaviors. Furthermore, the level-1 agent does not expect to receive any messages either because it thinks that others are not modeling others intentionally, so level-0 agents would determine there is no benefit to sending messages (especially in costly communication channels). Reasoning inductively, higher-level agents are also unable to reason the benefits of sending messages.

Gmytrasiewicz [2020] also notes this challenge, and to address it, treats level-0 agents as both "literal listeners" and "literal speakers", respectively. Level-0 agents act as literal listeners by incorporating any received information, though they do not attribute the sender as having been intentional (and thus honest or dishonest) in their communication. For

this process, Gmytrasiewicz proposes a separate update process than the Bayesian update defined previously in Eq. 1 that instead mixes the existing belief with new information. Likewise, level-0 agents act as literal speakers by sending messages to "no one in particular", optimistically assuming that other agents could take advantage of the communicated message. For this process, Gmytrasiewicz proposes a message generation function: with a probability $\alpha$, the agent communicates the honest marginal over its belief, and it does not communicate with probability $1 - \alpha$. This approach is better suited for agents that communicate belief marginals. As we intend to change the message primitives under agent openness, we model level-0 agents differently.

In particular, let $f_j : m_{i \leftarrow j} \rightarrow a_j$, which maps a message received from agent $j$ to its action $a_j \in A_j$, replace the fixed distribution ascribed by the level-0 agent $i$ to sender $j$'s actions. For more than one other agent, denote the vector of maps, one for each other agent, as $\mathbf{f_{-i}}$. Then, level-0 $i$'s updated belief about the environment state is:

$$
\begin{aligned}
b_{i,0}^t(s^t) = {} & \alpha \, O_i(s^t, a_i^{t-1}, \mathbf{f_{-i}}(\mathbf{m_{i \leftarrow -i}^t}), o_i^t) \\
& \times \sum_{s^{t-1}} b_{i,0}^{t-1}(s^{t-1}) \, T_i(s^{t-1}, a_i^{t-1}, \mathbf{f_{-i}}(\mathbf{m_{i \leftarrow -i}^t}), s^t) \quad (6)
\end{aligned}
$$

This update is analogous to a POMDP belief update with the modification that it allows messages received from others in the neighborhood to impact the updated belief. Consequently, given that the level-1 agent is aware that an agent modeled at level-0 updates its belief using Eq. 6, the level-1 agent may conclude that there is value in communicating with others because messages sent by it and received by others may indeed impact their beliefs over the state, which in turn may potentially affect their action choice.

Furthermore, let the level-1 agent in the open agent CI-POMDP consider an adapted version of the literal sender of Gmytrasiewicz [2020]. It stochastically generates a message that is honest about its presence or absence with a probability $\alpha$, while the remaining probability mass is uniformly spread across sending an incorrect suppressant level or no message. We denote this generator with the function, $g_i^\alpha : presence_i \rightarrow m_{i \rightarrow -i}$.

With both of these changes, level-1 agents are now incentivized to both send and receive messages as they believe messages will be processed and sent by the lower level agents. By induction, all higher-level agents are also incentivized to communicate, enabling decision making that also reasons about communicative acts in open agent systems.

## 4 MCTS FOR CI-POMDPS

We present an online planning algorithm for the open agent CI-POMDP model called CI-POMCP-PF$_O$ that uses Monte Carlo Tree Search (MCTS) to calculate the subject agent's set of optimal actions $OPT(b_{i,l}^t)$ (Eq. 4) for its current

belief $b_{i,l}^t$ and the resulting policy $\pi(b_{i,l}^t)$. This algorithm is the first general planning algorithm for CI-POMDPs, and it offers several important and non-trivial improvements over the state-of-the-art I-POMCP$_O$ algorithm [Eck et al., 2020] for decision-theoretic planning in open environments:

1. CI-POMCP-PF$_O$ reasons about benefits and costs of communication to determine when and what the subject agent should communicate with its neighbors, as well as incorporates information from received messages into its beliefs about other agent's mental models.

2. CI-POMCP-PF$_O$ produces solutions to a full CI-POMDP model where other agents are modeled as also solving a CI-POMDP of the world. This improves over solving an IPOMDP-Lite model where the neighbors are instead modeled using the simpler Nested-MDP model.

3. CI-POMCP-PF$_O$ addresses the large branching factor due to reasoning about receiving messages from all neighbors by projecting weighted PFs during each trajectory MCTS, rather than a single particle at a time. This brings recent advancements in single agent planning [Garg et al., 2019, Thomas et al., 2020, Sunberg and Kochenderfer, 2017] to multiagent contexts.

4. CI-POMCP-PF$_O$ can run fully online, requiring no offline precomputed neighbor policies, although it can also make use of offline neighbor policies if available.

### 4.1 MONTE CARLO TREE SEARCH

The POMCP algorithm [Silver and Veness, 2010] is a canonical approach for MCTS in partially observable environments. It operates by constructing an AND-OR tree of alternating belief (OR) and action (AND) nodes (e.g., Fig. 5 in the supplementary material) by following the general process in Alg. 2 in the supplementary material. The root node signified by $\varepsilon$ represents the agent's current belief about the environment, stored as an unweighted particle filter. Over several trajectories, the POMCP iteratively samples a particle (i.e., state) from the root belief, then projects the particle down the tree. During each trajectory, an action is chosen to balance the exploration-exploitation tradeoff using the UCB-1 heuristic $\underset{a \in A}{\arg\max} \, Q(h, a) + \sqrt{\frac{\log n(h)}{n(ha)}}$, where $h$ represents a history of actions and observations since the root node (alternatively a path from the root of the tree to a unique belief node), $Q(h, a)$ is the Q-value of action $a$ for the belief reached by history $h$, $n(h)$ and $n(ha)$ are the number of trajectories that have reached the belief node at history $h$ and simulated action $a$, respectively. Next, the algorithm simulates taking the chosen action in the particle's current state to produce a next state, reward, and observation. The algorithm then appends the action and observation to the history $h$ and recurses on the next belief at the new history $hao$ with the next state $s'$ as the trajectory's particle. If a leaf of the tree is reached, then the algorithm instead performs a

rollout by randomly choosing and simulating actions for the remaining horizon, and the leaf is expanded by adding its children action nodes and their children belief nodes (one for each observation). Finally, the algorithm unrolls the recursion by returning the reward sum earned from the current belief node to the leaf reached, updating $Q(h, a)$ using a rolling average, and incrementing $n(h)$ and $n(ha)$.

Once all trajectories have been exhausted, $OPT(b_{i,l}^t)$ is the child action(s) of the root node with the maximal $Q$ value. The agent physically acts by choosing an action from $OPT$ (e.g., uniformly at random), then follows the branch in the tree for the received observation to identify its next belief.

The I-POMCP$_O$ algorithm [Eck et al., 2020] extends POMCP to multiagent settings, where the subject agent solves an IPOMDP-Lite model, and addresses agent openness by maintaining beliefs about the presence of other agents within their mental models (Sec. 3.2). I-POMCP$_O$ is the first online planning algorithm for the I-POMDP family of models and demonstrated scalability to many-agent systems. Primary differences between I-POMCP$_O$ and single agent POMCP are that (1) particles contain not only an environment state, but also the subject agent's own presence state $present_i$ and a mental model for each neighbor[3] (that include $present_j$ states to model openness), and (2) the algorithm predicts (using precomputed level $l-1$ offline policies) the actions of the neighbors based on its mental models to simulate the next state, reward, and observation.

## 4.2 CI-POMCP-PF

Our novel algorithm CI-POMCP-PF$_O$ (Alg. 1) extends I-POMCP$_O$ to reason about not only physical actions, but also communicative actions that can enhance the agents' modeling of each other's presence in the open agent system. However, planning for the open agent CI-POMDP, rather than an IPOMDP-Lite that cannot reason about communication, requires overcoming several critical challenges. We first highlight these challenges and our approaches to addressing them, followed by an illustration of how planning occurs with our CI-POMCP-PF$_O$ algorithm.

First, the structure of the tree must be adapted to decide what to communicate, as well as incorporating received messages into next beliefs, illustrated in Fig. 1. The fanout of action nodes under each belief node increases linearly with the number of messages $|M|$ since now the agent must decide not only what action to perform in each situation, but also which message it will send (or not send a message at

---

[3]I-POMCP$_O$ achieves scalability in number of agents by selectively modeling only a subset of neighbors and extrapolating their behaviors to the collective system, relying on frame-action anonymity [Sonu et al., 2015, 2017]. We anticipate communication to be more critical in systems with a small number of agents and leave many-agent extensions to future work.

---

**Algorithm 1** CI-POMCP-PF

1: **procedure** CREATECOMMPLAN($b_{i,l}, \mathbf{m_{i \leftarrow -i}}$)
2:   **for** $traj \in 1, 2, \dots, \tau$ **do**
3:     UpdateTree($b_{i,l}, \mathbf{m_{i \leftarrow -i}}, 0, \varepsilon$)
4:   **return** $\underset{a \in A_i, m \in M}{\operatorname{argmax}} Q(\varepsilon, a, m)$

5: **procedure** UPDATETREE($b_{i,l}, \mathbf{m_{i \leftarrow -i}}, t, h$)
6:   **if** $t \geq H$ **then**
7:     **return** 0
8:   **if** $h$ is a leaf **then**
9:     Expand($h, i, l$)
10:     **return** Rollout($b_{i,l}, t$)
11:   $a, m_{i \rightarrow -i} \leftarrow$ ChooseActionComm($h$)
12:   $b'_{i,l}, \mathbf{m'_{i \leftarrow -i}}, r, o \leftarrow$ SimulateComm($b_{i,l}, \mathbf{m_{i \leftarrow -i}}, a, m_{i \rightarrow -i}$)
13:   $R \leftarrow r + \gamma *$ UpdateTree($b'_{i,l}, \mathbf{m'_{i \leftarrow -i}}, t + 1, hao\mathbf{m'_{i \leftarrow -i}}$)
14:   StoreResults($h, b_{i,l}, a, R$)
15:   **return** $R$

16: **procedure** SIMULATECOMM($b_{i,l}, \mathbf{m_{i \leftarrow -i}}, a, m_{i \rightarrow -i}$)
17:   $R \leftarrow 0$
18:   $\omega(o_i) \leftarrow 0, \quad \mu(o_i) \leftarrow \emptyset, \quad b_{i,l}^{o_i} \leftarrow \emptyset \ \forall o_i \in \Omega_i$
19:   **for** $(w, s, present_i, \times_{j \in N(i)} \langle b_{j,l-1}, present_j, \theta_j \rangle) \in b_{i,l}$ **do**
20:     **if** $l > 0$ **then**
21:       $a_j, m_{j \rightarrow j} \leftarrow$ CreateCommPlan($b_{j,l-1}, \mathbf{m_{j \leftarrow -j}}$) $\forall j$
22:     **else**
23:       $a_j \leftarrow f_i(m_{i \leftarrow j}), \quad m_{j \rightarrow j} \leftarrow g_j^\alpha(present_j) \ \forall j \in N(i)$
24:     $s', r, o_i, \mathbf{present'} \leftarrow$ Simulate($s, \mathbf{present}, a, \mathbf{a_{-i}}$)
25:     $w' \leftarrow w \cdot T_i(s, a, \mathbf{a_{-i}}, s') \cdot O(s', a, \mathbf{a_{-i}}, o_i)$
26:     $\omega(o_i) \overset{+}{\leftarrow} w', \quad \mu(o_i) \overset{\cup}{\leftarrow} (w', \mathbf{m'_{i \leftarrow -i}}), \quad R \overset{+}{\leftarrow} w \cdot r$
27:     $b_{i,l}^{o_i} \overset{\cup}{\leftarrow} (w', s', present_i', \times_{j \in N(i)} \langle b'_{j,l-1}, present_j', \theta_j \rangle)$
28:   $o_i \sim \omega(o_i), \quad \mathbf{m'_{i \leftarrow i}} \sim \mu(o_i)$
29:   **return** $b_{i,l}^{o_i}, \mathbf{m'_{i \leftarrow i}}, R, o_i$

30: **procedure** EXPAND($h, i, l$)
31:   $B_{i,l}(h) \leftarrow \emptyset, \quad n_{i,l}(h) \leftarrow 0$
32:   $n_{i,l}(ha) \leftarrow 0, \quad Q_{i,l}(h, a, m) \ \forall a \in A_i, m \in M$

33: **procedure** CHOOSEACTIONCOMM($h$)
34:   **return** $\underset{a \in A, m \in M}{\operatorname{argmax}} Q_{i,l}(h, a, m) + \sqrt{\frac{\log n_{i,l}(h)}{n_{i,l}(ham)}}$

35: **procedure** STORERESULTS($h, b_{i,l}, a, m, R$)
36:   $B_{i,l}(h) \leftarrow$ norm($B_{i,l}(h) \cup b_{i,l}$)
37:   $n_{i,l}(h) \overset{+}{\leftarrow} 1, \quad n_{i,l}(ham) \overset{+}{\leftarrow} 1$
38:   $Q_{i,l}(h, a, m) \overset{+}{\leftarrow} \frac{R - Q_{i,l}(h, a)}{n_{i,l}(ham)}$

---

all). Moreover, the fanout of belief nodes under each action node increases *exponentially* in the number of neighbors $|N(i)|$ as the subject agent $i$ must consider not only what observation it receives from the environment, but also what *combination* of messages it receives from all neighbors.

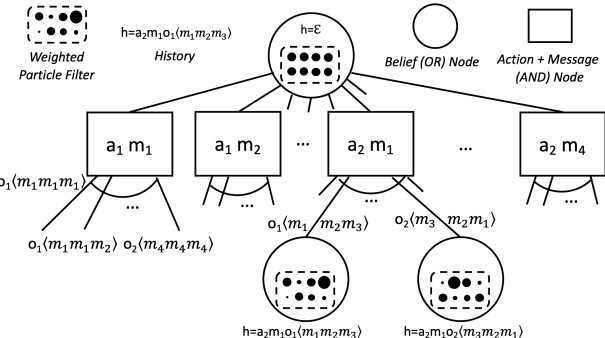

Figure 1: Example AND-OR tree created by CI-POMCP-PF with 3 neighbors, 2 actions, 4 messages, 2 observations. The fanout is 8 actions + message nodes and 128 belief nodes.

This high fanout is challenging due to its impact on the size of the tree, and it has severe implications from only projecting a single particle per trajectory [Sunberg and Kochenderfer, 2017]: almost all belief nodes near the bottom of the tree would suffer from particle deprivation because they would be reached only once with a single particle, hence their $Q$ estimates would be poor approximations – close to those estimated by loose bound QMDP [Littman et al., 1995]. Furthermore, high fanout implies that the leaves are shallower for a fixed number of trajectories compared to planning without communication, so the poorly approximated leaves will be near the root, and consequently $Q$ estimates will be poor not only at the leaves of the tree, but all throughout.

To address this first challenge, we adopt a recent technique used in multiple single-agent MCTS algorithms such as PFT-DPW [Sunberg and Kochenderfer, 2017], DESPOT-$\alpha$ [Garg et al., 2019], and $\rho$-POMCP [Thomas et al., 2020] that improve MCTS in environments with large observation spaces such as ours where received messages are also treated analogously to observations. To avoid poor $Q$ estimates due to particle deprivation, these algorithms instead employ a *weighted* particle filter and project the entire filter down the tree during each trajectory so that $Q$ estimates for each action node are obtained from more than a single particle leading to better approximations. A second benefit to this approach is that the agent's belief update after taking a physical action is more informed as the corresponding weighted particle filter in the second level of the tree will not suffer from particle deprivation, either.

The second challenge is that precomputing offline policies for the neighbors might not be tractable when (1) the mental models of neighbors are unknown until the agent operates in the environment (e.g., when different organizations contribute robots in response to a rapidly emerging wildfire), or (2) when the problem size is too large to afford planning for all possible scenarios (including the resulting beliefs from all possible combinations of received messages from all neighbors, which potentially exponentially expands the

number of reachable beliefs from the initial belief). To address this challenge, our CI-POMCP-PF$_O$ algorithm can operate fully online, predicting the behaviors of neighbors by *embedding* MCTS at one lower level each time a predicted action is needed for each neighbor. On the other hand, line 21 of Alg. 1 can also be replaced by a lookup from precomputed policies if available.

The online planning with CI-POMCP-PF$_O$ proceeds as follows. Each time the planning agent needs to choose an action, it constructs an AND-OR tree via $\tau$ Monte Carlo simulations using the recursive UpdateTree procedure. Each simulation starts at the root of the tree representing the agent's current belief about the environment state, presence of neighbors, and their nested beliefs. The agent simulates an action to perform and message to send sampled using the UCB-1 heuristic (e.g., which fire to fight and suppressant message to send) using the ChooseActionComm procedure.

It then calls the SimulateComm procedure to (1) simulate the reasoning of its neighbors at level $l - 1$ using the same CI-POMCP-PF$_O$ algorithm to predict their actions and the messages it will receive in the next time step, (2) simulate how the environment changes (e.g., new fire intensities and rewards received) based on everyone's chosen actions, (3) sample an observation about the environment state (e.g., how it sees the fires change) and received messages from neighbors (e.g., their communicated suppressant levels), and (4) propagate the particles in the agent's weighted particle filter. The sampled observation and received messages are taken from distributions $\omega$ and $\mu$ constructed during simulations based on the agent's weighted particle filter belief.

The UpdateTree procedure then follows the branches for the simulated action and sent message, as well as received observations and messages and either recursively repeats until the end of the planning horizon $H$ or until a leaf node is reached, at which it performs a rollout as in standard MCTS. Across all $\tau$ Monte Carlo simulations, the agent's planning tree (and hence policy) is refined. Finally, $OPT$ (Eq. 4) is calculated and an optimal action(s) and sent message(s) returned as the policy for the current belief.

## 5 EXPERIMENTS

We evaluate the CI-POMCP$_O$ algorithm (Alg. 1) on the wildfire suppression problem, a challenging benchmark for planning in open agent systems established previously [Chandrasekaran et al., 2016, Eck et al., 2020].[4]

**Setups** Agents are tasked with putting out fires of different sizes in the absence of prior coordination. Putting out small, medium, large, and huge fires provide agents shared rewards of 20, 50, 125, and 300, respectively, whereas a fire burning

---

[4]The source code for our implementation is available at https://github.com/OberlinAI/CommunicativeOASYS

out earns a shared penalty. The spread of fires is modeled on the dynamics of real wildfires Boychuk et al. [2009], Ure et al. [2015]; fires can take on five levels between non-existant to burned out. Agents have limited amounts of suppressant with *present* levels starting at full, followed by half full, then empty that transition stochastically when the agents take actions to fight adjacent fires or recharge while taking a NOOP action when empty (suppressant level reduces 25% and recharges to full with 50% probability). Details about agent and fire types are presented in Fig. 2.

We consider three different setups, illustrated in Fig. 2, which vary in the need for coordinated behavior. Setup 1 represents a situation where two agents each have unique small fires they can put out individually, as well as a shared fire that requires both agents to act simultaneously to reduce. Thus, agents can act independently, but they earn more rewards by acting together; here communication can help them coordinate. Setup 2 represents a more complicated scenario where no fires can be reduced independently, necessitating more coordination; instead, two fires require two agents to work together and a third fire requires all three agents to act simultaneously. Finally, Setup 3 further increases the size of fires and adds mixed types to determine how agents communicate within and across frames.

**Evaluations**   Within each setup, we evaluate the benefits of reasoning about communication by comparing our CI-POMCP-PF$_O$ algorithm with an ablated I-POMCP-PF$_O$ that keeps all extensions from state-of-the-art I-POMCP$_O$ [Eck et al., 2020], except that agents do not send or receive messages. We do not compare against other communication algorithms as no prior methods exist for the CI-POMDP framework. To further evaluate how well our model and algorithm reason about balancing the trade offs between the benefits and costs of communication, we let communication costs for each sent message assume a cost in $\{0, 0.05, 0.1, 0.2, 0.5, 1\}$. Agent performance is measured by the average rewards, and we also evaluate how many messages were sent, also when and how messages were sent, as communication costs vary. All results are averaged over 50 runs, and each run takes 15 time steps.

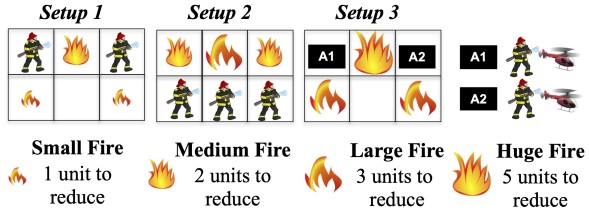

**Setup 1**   **Setup 2**   **Setup 3**

**Small Fire**
1 unit to reduce

**Medium Fire**
2 units to reduce

**Large Fire**
3 units to reduce

**Huge Fire**
5 units to reduce

Figure 2: Our setups involve varying numbers, sizes, and positions of fires, as well as varying numbers of agents and their types. Different types of fires require different units of suppressant to reduce. Ground firefighters apply 1 unit while helicopters bring 2 units to the firefighting.

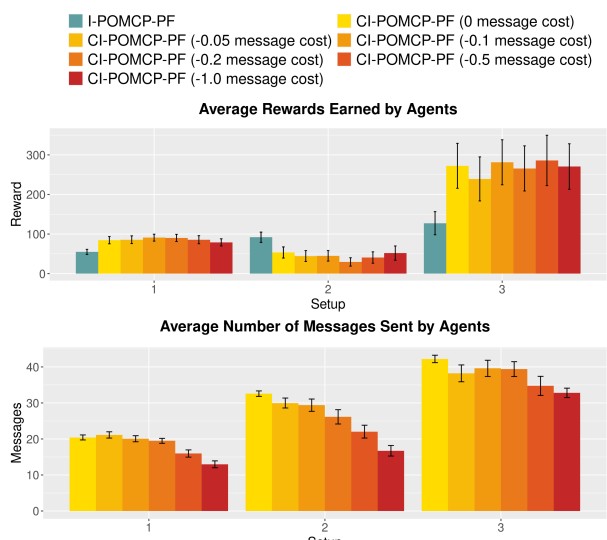

Figure 3: Average rewards and messages sent per agent. Error bars = 95% confidence intervals

**Hyperparameters**   The $f$-function assumes that agents with full suppressant will be around long enough to fight the largest fire in the environment, whereas agents with half suppressant will need to leave soon and favor smaller fires; empty suppressant maps to NOOP and no message maps uniformly to all actions. The $g$-function assumes level-0 agents are honest with $\alpha = 95\%$. Agents plan with $\tau = 1000$ trajectories at level $l = 1$, horizon $H = 10$, 50 particles in the particle filter, and $c = 50, 75, 100$ for Setups 1, 2, and 3. Neighbors' level-0 policies were precomputed.

**Results**   From Fig. 3, we first observe that in both wildfire Setups 1 and 3, agents planning with CI-POMCP-PF$_O$ produced policies that earned *statistically significantly* greater rewards than I-POMCP-PF$_O$. In Setup 1, I-POMCP-PF$_O$ agents focused first on the small fires they could handle individually, then worked on the shared fire when the small fires were put out, whereas communicating CI-POMCP-PF$_O$ agents focused first on the shared fire that required coordination, which was enabled through information shared in messages, to earn higher rewards and better task accomplishment. We observed similar behavior in Setup 3, where non-communicating agents focused on the smaller fires first, while CI-POMCP-PF$_O$-enabled coordination to put out the huge fire worth the most rewards. Overall, our model and algorithm reasoned successfully about *when* to communicate.

On the other hand, we observe for Setup 2 that I-POMCP-PF$_O$ statistically significantly outperformed the CI-POMCP-PF$_O$ agents. Once again, I-POMCP-PF$_O$ agents without communication focused their initial attention on the smaller fires; the fire reduced depended on the initial choice of the middle agent (who equally decided between both small fires). With communication, the

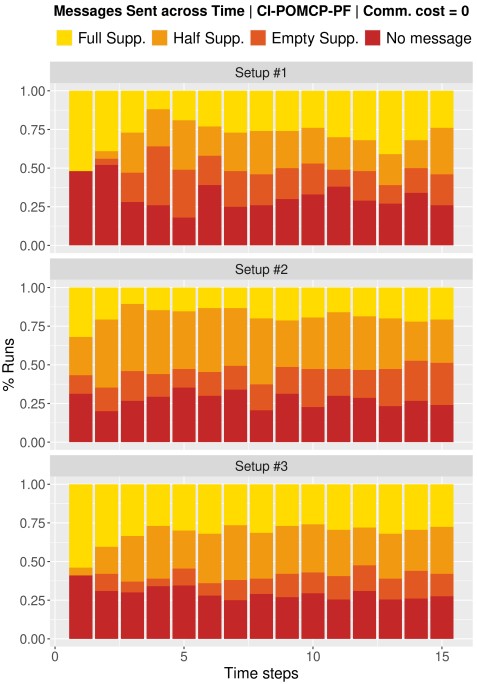

Figure 4: Messages sent in Setups 1-3 (cost = 0)

CI-POMCP-PF$_O$ encouraged more agents to attempt to put out the large fire initially. However, because the large fire requires all three agents to be present and work together, they failed whenever one of them ran out of suppressant before the fire was put out, or someone chose to switch actions (possibly due to randomness during MCTS planning). Thus, they were delayed in shifting to the smaller fires, which instead burned out and led to a penalty and not a reward. Overall, communication led to more coordinated behavior, but the challenge of needing all agents to put out the shared fire in this particular setup was too difficult to consistently overcome in an open environment.

Across all setups, we also observe that as communication costs increased, CI-POMCP-PF$_O$ agents communicated less frequently, yet their rewards earned did not significantly decrease. This implies that in response to communication costs of varying levels, CI-POMCP-PF$_O$ enables agents to choose *when* it is best to communicate (especially in Setups 1 and 3) to balance the benefits of communicating when it is most effective against the costs of communication. In other words, the agents became more efficient when they utilized communication to improve coordination and rewards received as the communication cost increased.

Finally, we investigate the messages sent by agents in Fig. 4 (c.f., Appendix C in the supplementary material for results with cost = 1 and for individual agents in each setup). In Setups 1 and 3, all agents frequently communicated a *full suppressant* message as they began fighting fires (approximately 50% of messages; higher costs increased *no message*

frequency), which corresponds both to the starting suppressant levels of agents (the sent messages were honest) and the message that maps to fighting the shared fire (the agent's true action) in the $f$-function. As the agents continued operating, their messages correspond more closely to the actions chosen (c.f., Appendix B in the supplementary material) than their true suppressant levels. This implies that level-1 agents determined that they could indeed influence the behaviors of level-0 literal listening neighbors and sent messages they believed would be interpreted in such a way that would lead to coordinated actions. In Setup 2, agents also sent messages often corresponding to their actions, but the challenge of at least one agent running out of suppressant before putting out the shared fire still limited their task success.

# 6 CONCLUDING REMARKS

Real-world domains often exhibit agent openness, where agents may leave the system and then possibly return. We presented an extension to the recent CI-POMDP framework to allow the agents to decide when and what to communicate about their presence in the system to provide information about their availability, thereby allowing agents to better infer their neighbors' unobserved presence. We presented the CI-POMCP-PF$_O$ algorithm, a MCTS-based online planning algorithm that extends the state-of-the-art I-POMCP$_O$ to enable reasoning about communication in open and typed multiagent systems. Simulations in three challenging scenarios of the wildfire suppression benchmark demonstrated that the novel algorithm not only enables agents to reason about *what* to communicate in order to produce better coordination and task accomplishment, but also *when* to communicate to balance the benefits and costs of the communicative acts.

We restricted our attention to open environments where existing agents may exit and re-enter the system, but the approach could also be extended to address environments where *new agents* join after the task begins. The subject agent would first need to infer the presence of new agent(s), and then add a new mental model for this neighbor, which would allow the agent to reason about the new neighbor's behavior as it plans. Utilizing online planning like the CI-POMCP$_O$ algorithm enables such adaptive changes to planning in the complex environment. Of course, an unanswered question is how the subject agent will become aware of such new neighbors, especially when communication is sparse, which we plan to investigate as future work, along with adapting our approach to many-agent open environments.

### Acknowledgements

This research was supported by a collaborative NSF grant #IIS-1909513 (to AE), #IIS-1910037 (to PD), and #IIS-1910156 (to LKS). We thank the anonymous reviewers for their valuable feedback.

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
