# OpenReview forum: "Decision-Theoretic Planning with Communication in Open Multiagent Systems"
_auai.org/UAI/2022/Conference — UAI 2022 Poster_

### Official Review · Reviewer_Pq8Y · 2022-04-09

**Q2(1) Originality/Novelty:** 2
**Q2(2) Significance/Impact:** 2
**Q2(3) Correctness/Technical Quality:** 3
**Q2(6) Clarity Of Writing:** 4
**Q6 Overall Score:** 4
**Q8 Confidence In Your Score:** 3

**Q1 Summary And Contributions:**

The paper extends the recently-developed communicative interactive POMDP framework to model agents that can join or leave the system, and extends a Monte-Carlo approach to solving POMDPs to allow planning using this extended model. The approach is illustrated by experiments in a simulated multi-agent fire-fighting domain.


**Q2 Assessment Of The Paper:**

More detailed information regarding each of these aspects is given below:

**Q2(4) Quality Of Experiments (Optional):**

3: Good: The experimental evaluation is adequate, and the results convincingly support the main claims.

**Q2(5) Reproducibility:**

3: Good: Key resources (e.g., proofs, code, data) are available and key details (e.g., proofs, experimental setup) are sufficiently well-described for competent researchers to confidently reproduce the main results.

**Q3 Main Strengths:**

Technically sound and clearly-written paper that describes some enhancements of previous work in this area.

The two contributions are (1) an enhancement of a previously-developed model of multi-agent POMDPs with communicative actions so that it also models of the presence of absence of agents in an *open* multi-agent system, and (2) an extension of a Monte-Carlo tree search algorithm for multi-agent POMDPs that allows planning in this extended model.

**Q4 Main Weakness:**

Although technically sound, both contributions (enhanced model and enhanced algorithm) seem to be straightforward and incremental enhancements of a previously-developed model and algorithm, and thus the degree of originality/significance seems modest. For example, I looked at the Eck et al. paper from AAAI-20 that is referenced in this paper, and there seem to be as many similarities as differences between the papers.

**Q5 Detailed Comments To The Authors:**

Although the enhancements described in this paper are obviously sound, I will mention that I have some reservations about the value of creating increasingly ornate models. As pointed out in the third paragraph of section 3, an open system can be more simply modeled by a state variable that represents whether an agent is present or not, and I read but did not quite follow the explanation for not using this simpler approach. In fact, I’m not convinced that communicative actions cannot be more simply modeled by performing regular state-changing actions on a variable that is observed by the communicating agents. In summary, some enhancements of a model make it possible to do things the original model cannot do, and others are "syntactic sugar" that make the model more convenient to use, but not necessarily more powerful. I suspect these enhancements are of the latter type, but would like to be convinced otherwise. If the simpler model is inadequate for modeling an open multi-agent system, for one reason or another, it may help to introduce an example early in the paper that more clearly motivates and explains why these enhancements of the model are necessary.

**Q7 Justification For Your Score:**

A well-written and technically sound paper that makes an incremental contribution of perhaps modest significance.

**Q9 Complying With Reviewing Instructions:**

1: Yes.

---

### Official Review · Reviewer_KEbY · 2022-04-11

**Q2(1) Originality/Novelty:** 2
**Q2(2) Significance/Impact:** 2
**Q2(3) Correctness/Technical Quality:** 3
**Q2(6) Clarity Of Writing:** 4
**Q6 Overall Score:** 7
**Q8 Confidence In Your Score:** 3

**Q1 Summary And Contributions:**

A decision-theoretic framework for communicating among agents in an open multi-agent system that balances the tradeoff between the overhead of communication with the enhanced ability to coordinate.

**Q2 Assessment Of The Paper:**

More detailed information regarding each of these aspects is given below:

**Q2(4) Quality Of Experiments (Optional):**

3: Good: The experimental evaluation is adequate, and the results convincingly support the main claims.

**Q2(5) Reproducibility:**

3: Good: Key resources (e.g., proofs, code, data) are available and key details (e.g., proofs, experimental setup) are sufficiently well-described for competent researchers to confidently reproduce the main results.

**Q3 Main Strengths:**

The paper contributes new data about the relative costs and benefits of inter-agent communication.

The paper might offer developers of open multi agent systems new strategies for coordination

The paper seems correct in its technical results.

The paper offers real world-based experiments to evaluate their system

The paper is well written and complete.

The motivating application (fire fighting robots) is timely and illustrative of the problem they are trying to solve.

**Q4 Main Weakness:**

The paper is an extension to an existing approach and it’s not clear how non-trivial the new ideas are. In other words, it seems like a minor tweak to an existing framework.

Not convinced the impact will be broadly felt in the community.



**Q5 Detailed Comments To The Authors:**

Section 3.2 seems to drag the paper down with unnecessary detail. It seems to show that nested agent modeling and communication are incompatible, which to me is a flaw in the overall agent model, and the solution presented seems to be somewhat of a hack. There is nothing deeply interesting about the problem being addressed in this section, or its fix, and I would recommend trying to drawing attention to the problem in a single paragraph.

The graphs in Appendix C, although it adds to completeness, does not seem to me to be required. The qualitative summary in the last paragraph in section 5 is probably all you need here.

**Q7 Justification For Your Score:**

It was well written, well motivated with a real world scenario, the experiments clearly demonstrated and explained the advantages of their approach.

The work does not seem to be terribly original or game-changing with respect to state of the art. Although the results are promising, it's not clear that the approach  will be broadly applicable by the multi-agent community.

**Q9 Complying With Reviewing Instructions:**

1: Yes.

---

### Official Review · Reviewer_xiSK · 2022-04-12

**Q2(1) Originality/Novelty:** 3
**Q2(2) Significance/Impact:** 3
**Q2(3) Correctness/Technical Quality:** 3
**Q2(6) Clarity Of Writing:** 3
**Q6 Overall Score:** 8
**Q8 Confidence In Your Score:** 4

**Q1 Summary And Contributions:**

The paper puts forward a method for decision-theoretic planning enabling agents to plan with communicative actions in open multiagent systems. An algorithm optimised for large-observation spaces is proposed, and then tested on the wildfire suppression benchmark.

**Q2 Assessment Of The Paper:**

More detailed information regarding each of these aspects is given below:

**Q2(4) Quality Of Experiments (Optional):**

3: Good: The experimental evaluation is adequate, and the results convincingly support the main claims.

**Q2(5) Reproducibility:**

3: Good: Key resources (e.g., proofs, code, data) are available and key details (e.g., proofs, experimental setup) are sufficiently well-described for competent researchers to confidently reproduce the main results.

**Q3 Main Strengths:**

Overall, the content is interesting and relevant to the conference. I've found the writing very clear to follow and effective in guiding the reader from the background notions to the novel aspects. The idea to integrate the communication acts in the planned strategy seems very promising, as also confirmed by the experimental results.


**Q4 Main Weakness:**

No particular weaknesses

**Q5 Detailed Comments To The Authors:**

My only curiosity concerns this aspect: the second scenario is the only one in which the proposed algorithm did not outperform the baseline. The reasons for these results are expressed clearly: agents' tendency to prefer cooperation instead of individual actions exposes them to unpredictable events (suppressant shortage). Anyway, I would have also expected a discussion on the possible solutions to this problem. For example, changing the simulation hyperparameters has an impact on the results? Do the authors think that changing the set of available communicative messages could help?

**Q7 Justification For Your Score:**

The paper is good.

**Q9 Complying With Reviewing Instructions:**

1: Yes.

---

### Official Review · Reviewer_dNs8 · 2022-04-15

**Q2(1) Originality/Novelty:** 2
**Q2(2) Significance/Impact:** 2
**Q2(3) Correctness/Technical Quality:** 3
**Q2(6) Clarity Of Writing:** 3
**Q6 Overall Score:** 6
**Q8 Confidence In Your Score:** 3

**Q1 Summary And Contributions:**

The paper addresses decision-theoretic planning in open multiagent systems in presence of explicit communication between the agents.
In particular, the paper proposes a variant of the CI-POMDP model that accounts for openness and an algorithm for calculating the optimal actions of an agent in such a model.

**Q2 Assessment Of The Paper:**

More detailed information regarding each of these aspects is given below:

**Q2(4) Quality Of Experiments (Optional):**

3: Good: The experimental evaluation is adequate, and the results convincingly support the main claims.

**Q2(5) Reproducibility:**

2: Fair: Key resources (e.g., proofs, code, data) are unavailable but key details (e.g., proof sketches, experimental setup) are sufficiently well-described for an expert to confidently reproduce the main results.

**Q3 Main Strengths:**

The proposed approach is potentially interesting and has been shown to work well in an experimental setting.

**Q4 Main Weakness:**

The proposed model and algorithms are novel but quite incremental with respect to the state of the art.
The paper is rather clear, but more details (e.g., about algorithms and experiments) could be provided.

**Q5 Detailed Comments To The Authors:**

The form of openness considered in the paper is rather restrictive: agents can only leave and rejoin the system. The most difficult case in which a previously unknown agent joins the system is not considered.
How could the proposed model be modified in order to address also this case?

In Section 2, some intuition about the fundamental notion of level-l interactive states should be provided to make the paper more self-contained. Moreover, a discussion on the possible values for l could be added (in the following, only examples with l=1 are made).

The purpose of Section 3 is not fully clear. It lengthly motivates the use of communication and the most relevant part appears to be the second half of Section 3.2. Perhaps some material could be removed and the section could be shortened.

Space saved (see the comment above) could be used to better explain the details of Algorithm 1.

Experiments show that the proposed approach works in the wildfire suppression setting.
However, in setup #2, the algorithm I-POMCP-PF_{O} that does not use communication performs better. This is counter-intuitive: the CI-POMCP-PF_{O} algorithm can learn a superset of what the I-POMCP-PF_{O} can (since the former can simply learn to ignore messages), right? Does the problem lie in the incomplete learning process or in the type of messages that agents are exchanging? Maybe messages that are more related to the coordination between agents could help more the system?

Minor issues:
- Use parentheses (\citep{}) for references.
- Page 1: it likely
- Page 4: that optimize that maximize rewards
- References [Mirsky et al., 2022] and [Sunberg and Kochenderfer] are without venues, pages, ...

**Q7 Justification For Your Score:**

The paper provides an interesting but limited incremental contribution, which nevertheless appears to be sound.

**Q9 Complying With Reviewing Instructions:**

1: Yes.

---

### Decision · Program_Chairs · 2022-05-15

**Decision:**

Accept (Poster)

**Comment:**

Meta Review: The reviewers felt that this paper offered a valuable contribution to the area of multiagent planning, and specifically in the use of communication in interactive POMDPs to handle certain forms of “open systems” (involving the presence/absence of other agents). There was an appreciation for the novelty of the problem, the new algorithm proposed (an extension of MCT to handle the “open” setting), and the empirical evaluation. The reviewers raised some questions about the general novelty vs. incrementality of the approach, and made a number of suggestions that could greatly improve the presentation and emphasis in the paper. The author response clarified a number of questions and convinced the reviewers that paper revisions would be forthcoming that would enhance the impact of this work. The authors are strongly urged to make the revisions suggested (and discussed in their rebuttal).